# Myoepithelial progenitors as founder cells of hyperplastic human breast lesions upon *PIK3CA* transformation

Nadine Goldhammer [1,2], Jiyoung Kim [1,2], René Villadsen[1], Lone Rønnov-Jessen [3] & Ole William Petersen [1,2✉]

The myoepithelial (MEP) lineage of human breast comprises bipotent and multipotent progenitors in ducts and terminal duct lobular units (TDLUs). We here assess whether this heterogeneity impacts on oncogenic *PIK3CA* transformation. Single cell RNA sequencing (scRNA-seq) and multicolor imaging reveal that terminal ducts represent the most enriched source of cells with ductal MEP markers including α-smooth muscle actin (α-SMA), keratin K14, K17 and CD200. Furthermore, we find neighboring CD200[high] and CD200[low] progenitors within terminal ducts. When sorted and kept in ground state conditions, their CD200[low] and CD200[high] phenotypes are preserved. Upon differentiation, progenitors remain multipotent and bipotent, respectively. Immortalized progenitors are transduced with mutant *PIK3CA* on an shp53 background. Upon transplantation, CD200[low] MEP progenitors distinguish from CD200[high] by the formation of multilayered structures with a hyperplastic inner layer of luminal epithelial cells. We suggest a model with spatially distributed MEP progenitors as founder cells of biphasic breast lesions with implications for early detection and prevention strategies.

[1] Department of Cellular and Molecular Medicine, University of Copenhagen, Copenhagen N, Denmark. [2] Novo Nordisk Foundation Center for Stem Cell Biology, University of Copenhagen, Copenhagen N, Denmark. [3] Section for Cell Biology and Physiology, Department of Biology, University of Copenhagen, Copenhagen Ø, Denmark. ✉email: owp@sund.ku.dk

The presence of the so-called TDLUs at the most distal parts of a branching ductal system is a distinctive feature of the human breast gland. The TDLUs consist of varying amounts of small ductules or alveoli connected to a single terminal duct embedded in a loosely arranged lobular stroma. Terminal ducts drain into larger extralobular ducts, which eventually connect to the nipple. At a higher resolution, TDLUs and ducts are universally composed of an inner layer of luminal cells and an outer layer of MEP cells taken to represent the only two epithelial lineages present in the human breast (reviewed in[1]). The interest in the subtleties of human breast anatomy remains pertinent not least due to the ongoing quest for the breast cancer cells-of-origin. We and others have attempted to approach this based on cell culture of reduction mammoplasty-derived cells followed by immortalization and further oncogenic transformation[2–5]. Most studies narrow down to a luminal progenitor as the source of in particular the basal-like subtype of breast cancer[6,7]. Whether the MEP lineage plays an active role in tumor evolution aside from an aberrant interaction with luminal cells[8] is less well studied. In fact, among all the cell types in the breast, MEP cells are among the least analysed[9].

We recently demonstrated two different differentiation repertoires of MEP progenitors from TDLUs and ducts, respectively[10]. Whereas ductal MEP progenitors homogeneously differentiate into K19$^+$ cells, TDLU-derived MEP are multipotent and generate both K19$^+$ and K19$^-$ luminal cells as is also seen in an age-dependent manner in TDLUs in vivo[10]. With this in mind and being well aware that most breast cancers presumably arise in TDLUs[11,12], we here took on to further interrogate the consequences of the oncogenic transformation of MEP progenitors from a regional anatomical perspective. Since in mice, luminal estrogen receptor-positive (ER$^+$) tumors are readily formed by expression of oncogenic $PIK3CA^{H1047R}$ within the MEP lineage, and $PIK3CA$ is the most commonly found driver mutation and early event in human breast cancer[13–15] (reviewed in[16]), and since the MEP lineage shows a unique PI3K signature in breast lesions of known $PIK3CA$ genotype[17,18], we here decided to focus on the expression of this oncogene.

We show that serial passaging of myodifferentiated MEP progenitors characterized and isolated based on scRNA-seq and fluorescence-activated cell sorting (FACS) into CD200$^{low}$ and CD200$^{high}$ populations reveals inherently distinct luminal differentiation repertoires. Thus, only CD200$^{low}$ progenitors are able to form keratin K14$^-$/K17$^-$/K19$^+$ luminal cells, that is, the phenotypic equivalent to most breast cancer cells[19]. Further transduction of CD200$^{low}$ MEP progenitors with $PIK3CA^{H1047R}$ leads to transformation into biphasic lesions with luminal hyperplasia resting on a seemingly uninvolved layer of MEP cells. Taken together, our results are in favor of a cell-of-origin of biphasic lesions in MEP cells of the TDLU.

## Results

**Single-cell sequencing and multicolor imaging reveal CD200 as an eligible marker for separation of MEP progenitors.** We previously provided a FACS-based protocol for a rough distinction and separation of TDLU-derived and duct-derived MEP by use of podoplanin known to regulate mammary stemness via Wnt/beta-catenin signaling[10,20]. To further increase the resolution specifically at the level of the progenitors within the terminal ducts, we here combined targeted micro-collection of TDLUs and ducts and scRNA-seq analysis of cells derived from three different reduction mammoplasties with multicolor imaging of normal breast MEP cells. TDLUs and ducts were trypsinized, and TDLU- and duct-derived epithelial single cells were sorted in a FACS-based protocol with Trophoblast cell surface antigen 2 (Trop2)

and p75 Neurotrophin Receptor p75NTR/CD271[10,21] (Fig. 1a). Equal numbers of Trop2$^+$/CD271$^{high}$ TDLU and ductal cells at the entry point were submitted to scRNA-seq. Unsupervised clustering using Seurat (version 3)[22,23] of 18,678 cells representing three biopsies, revealed that the cells could be divided into ten distinct clusters, 0-9 (Supplementary Tables 1 and 2), which were located in close proximity to each other in the t-distributed stochastic neighbor embedding (t-SNE) plot (Fig. 1a and Supplementary Fig. 1a). Very few cells segregated into clusters 7, 8, and 9 (Supplementary Table 1). By comparing differentially expressed genes (DEGs) in clusters 8 and 9 with gene sets in the Molecular Signatures Database (https://www.gsea-msigb.org/gsea/msigdb/index.jsp), cells in these clusters were identified as contaminating mesenchymal and vascular cells, respectively (Supplementary Fig. 1b and Supplementary Data 1). Cluster 7 comprised contaminating luminal epithelial cells as determined by the expression of known luminal marker genes (Supplementary Fig. 1c and Supplementary Data 1). Therefore, only clusters 0–6 were subjected to further analysis (Supplementary Data 2). Processing of MEP cells derived from TDLUs and ducts in separate revealed a significantly higher contribution from TDLUs in clusters 3 and 4, and from ducts in clusters 2 and 5 (Fig. 1b). To further narrow down markers of TDLUs, we next analyzed DEGs that were up- or downregulated in cluster 3 taken to be enriched in TDLU-derived cells. In particular, Cluster of Differentiation 200 (CD200), Adhesion molecule with Ig like domain 2 (AMIGO2), and Family With Sequence Similarity 126 Member A (FAM126A) were positively identified in cluster 3, while conversely, Retinoic Acid Receptor Responder 1 (RARRES1), Thioredoxin Interacting Protein (TXNIP) and Tumor Necrosis Factor Superfamily Member 10 (TNFSF10) apparently were highly expressed in clusters 0 and 2, but not in cluster 3 (Fig. 1c). Among these, we were particularly intrigued by the expression pattern of CD200 and AMIGO2 as candidate tools for the subsequent sorting of specific MEP cell populations. Both markers came up as DEGs in cluster 3 after filtration with respect to surface markers as annotated in the human surfaceome[24] (Fig. 2a, Supplementary Data 2). A screen of antibodies revealed that CD200, but not AMIGO2, was suitable for both immunostaining and FACS. Accordingly, crude preparations of uncultured MEP were sorted with Trop2 and CD271, followed by gating into CD200$^{low}$ and CD200$^{high}$ cells (Fig. 2a), and, depending on the biopsy ($n = 9$), CD200$^{low}$ cells represented a separate population accounting for 5–15% of the MEP cells (Supplementary Fig. 2a). To elucidate whether the sorted CD200$^{low}$ and CD200$^{high}$ cells corresponded to clusters identified by scRNA-seq in terms of gene expression profiles, Real-Time quantitative Reverse Transcription PCR (RT-qPCR) was performed. While cluster 3 markers CD200, AMIGO2, and FAM126A were significantly higher expressed in CD200$^{high}$ cells, RARRES1, TXNIP, and TNFSF10 that were negatively correlated with cluster 3 were expressed at a significantly higher level in CD200$^{low}$ cells (Fig. 2a).

To better elucidate cluster identities, we mapped K14 and K17 as reference ductal markers[10]. Rather than appearing in cluster 2 and 5 as one would expect from the staining history with these markers and the relative ductal enrichment within these clusters, K14 and K17 expressing cells showed up primarily in the TDLU-enriched cluster 3 together with CD200$^{high}$ cells (Fig. 2b). This was confirmed for K17 by immunostaining. A significantly lower number of cells stained strongly for K17 among CD200$^{low}$ cells compared to CD200$^{high}$ cells (Fig. 2b). By multicolor imaging of K17 and CD200 (Fig. 2c) it became clear that cluster 3 most likely is enriched by K17$^+$/CD200$^+$ cells representing terminal ducts within the TDLUs. Accordingly, in a sample of biopsies, the expression of CD200, as determined by immunostaining, was higher in terminal ducts compared to alveoli in fifteen out of thirty

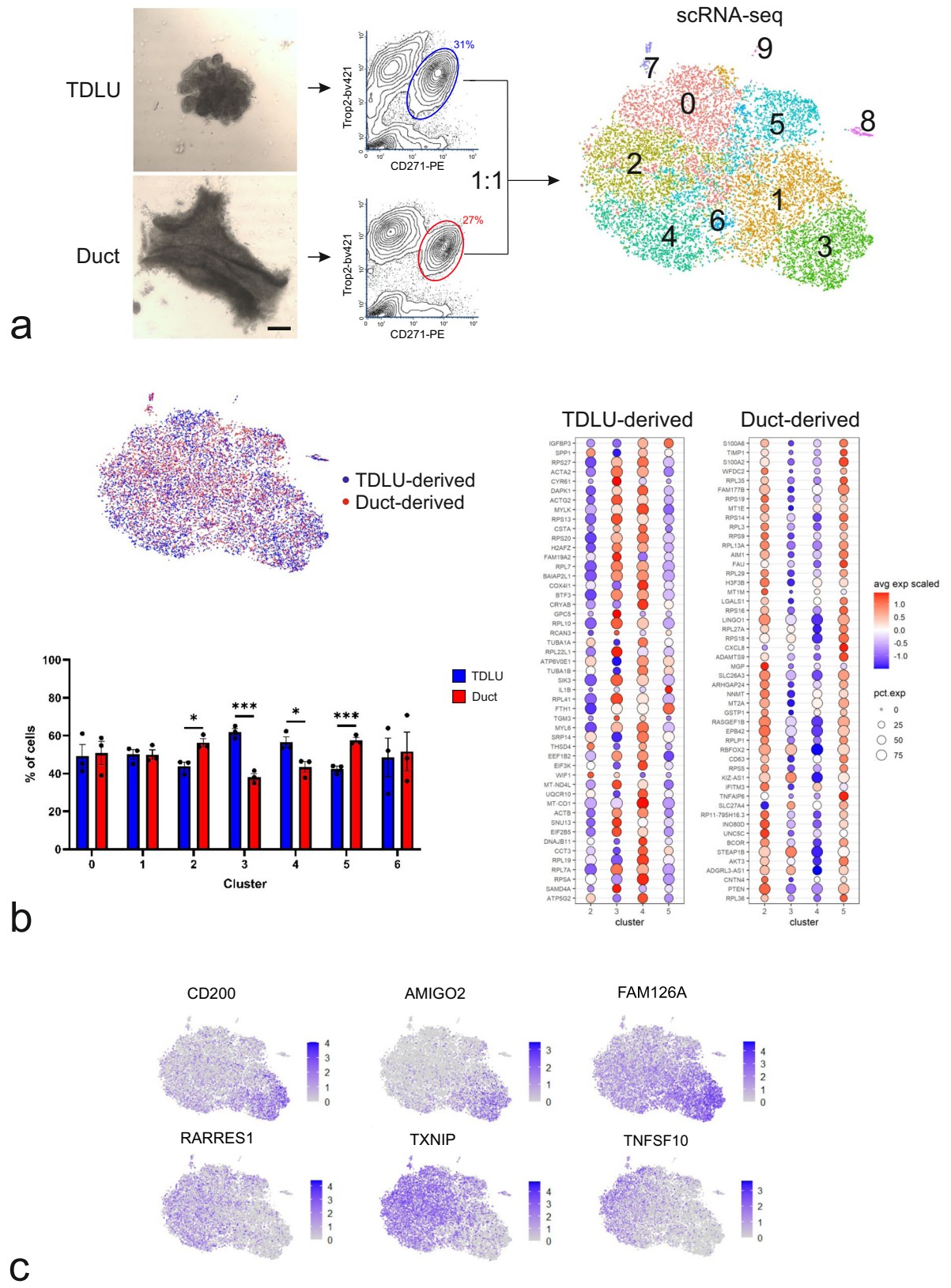

biopsies (Supplementary Fig. 2b). Intriguingly, both terminal ducts and ducts concomitantly harbored small foci of CD200$^{low}$ cells, which was most clearly revealed by multicolor imaging with the omnipresent MEP-specific SMA (Supplementary Fig. 2b). This defines three profiles of CD200 MEP cells for further scrutiny as summarized in Supplementary Fig. 2b: K17$^{-}$/CD200$^{low}$ alveolar cells, K17$^{+}$/CD200$^{low}$ terminal duct and ductal cells, and K17$^{+}$/ CD200$^{high}$ terminal duct and ductal cells. This striking spatial heterogeneity within the MEP lineage and the reproducible FACS separation offers a unique opportunity to interrogate a potential susceptibility to oncogenic transformation and a detailed resolution of the MEP progenitor compartment.

**Fig. 1 Spatial mapping of MEP progenitors at the single-cell level. a** (left) Phase-contrast micrographs of representative micro-collected TDLU and duct (scale bar, 200 µm). (middle) Contour FACS plots of single cells from trypsinized, micro-collected organoids labelled with Trop2 and CD271 for sorting of MEP cells (circles indicate approximate gates and percentages of isolated cells) derived from TDLU (blue) and ducts (red), respectively, which were submitted to scRNA-seq. (right) t-SNE plot of the integrated scRNA-seq profiles of 18,678 cells from three biopsies, subdivided by unsupervised clustering into clusters 0-9, indicated by separate coloring. **b** (left, upper) Same t-SNE plot as in a, colored to show MEP cells derived from TDLUs (blue) and ducts (red), respectively. (Left, lower) Bar graph depicting the contribution of TDLU- or duct-derived MEP cells to each cluster. Bars show mean ± standard error of mean (SEM) (*$p < 0.05$ and ***$p < 0.005$ by multiple unpaired t tests with Bonferroni correction, $n = 3$ biopsies). (right) Cluster-assigned bubble plots of the fifty most up-regulated TDLU-derived and duct-derived DEGs in clusters 2–5. Color indicates the average expression across cells within a cluster and the size indicates the percentage of expressing cells. **c** Same t-SNE plot as in A, colored according to genes expression levels from 0 (grey, low expression) up to 4 (purple, high expression). *CD200*, AMIGO2, and *FAM126A* are upregulated, while *RARRES1*, *TXNIP*, and *TNFSF10* are downregulated in cluster 3.

**CD200$^{low}$ MEP cells are multipotent progenitors and a facultative source of the luminal epithelial lineage.** Preservation of myodifferentiation is a prerequisite for the elucidation of MEP progenitor activity[25] and their potency to generate luminal cells in culture and in vivo[10]. With the aim of supporting a myo-differentiated phenotype for extended culture of CD200 cells, we therefore sought to optimize culture conditions favoring expression of the ultimate myodifferentiation marker, α-SMA protein. For this purpose, we tested interferon-alpha (IFNα), which has recently been shown to promote murine mammary stem cell activity[26], RepSox, originally shown to trap intermediate cell types during reprogramming of stem cells[27] and low oxygen, which contributes to the maintenance of stem cell potency[28]. When grown in Myo medium on human fibroblast feeders, crude preparations of MEP cells gradually lose their expression of α-SMA with passaging (Fig. 3a), and this outcome was not changed appreciably when the cells were exposed to either IFNα, RepSox or low oxygen. However, upon exposure to IFNα, RepSox, and low oxygen in combination (here termed Myo$^+$), a high level of α-SMA was achieved in primary culture as well as in subsequent passages (Fig. 3a), and maintained up to passage 5. The level of myodifferentiation depends on biopsy of origin, which in the optimal case includes more than 95% myodifferentiated cells in primary culture in Myo$^+$ (Supplementary Fig. 3a). This condition serves as ground state conditions to sustain myodifferentiated CD200$^{low}$ and CD200$^{high}$ K14$^+$/K19$^-$ progenitors (Fig. 3a, b and Supplementary Fig. 3b). To test for stem cell potential, the generation and differentiation of luminal cells from CD200$^{low}$ and CD200$^{high}$ ultra-pure progenitors were induced upon switch to culture without fibroblast feeders in MEGM[10] supplemented with Activin/NODAL/TGF-β pathway inhibitor, A83-01[29] (MEGM$^+$). While both populations within 10 days responded by the generation of K14$^+$/K19$^+$ progenitors, in addition, islets of more mature luminal K14$^-$/K19$^+$ cells appeared in the CD200$^{low}$ cultures (Fig. 3b, Supplementary Fig. 3b, and Supplementary Fig. 4a). It was evident that while CD200$^{low}$ and CD200$^{high}$ populations exhibited similar frequencies of α-SMA-positive and K14$^+$/K19$^+$ progenitors, generation of luminal K14$^-$/K19$^+$ cells was seen among CD200$^{low}$ cells only (38.0% +/− 12.6% versus 0.7% +/− 0.5%, Supplementary Fig. 4a). RT-qPCR was performed to detect regulation of MEP and luminal gene expression upon differentiation. While expression of *KRT14* and *TP63* was reduced and *EPCAM* was induced at similar levels in CD200$^{low}$ and CD200$^{high}$, *ELF5* and *KRT19* were induced to a significantly higher level in CD200$^{low}$ cells (Supplementary Fig. 4b). Finally, the culture of differentiated CD200$^{low}$ MEPs in a medium described to facilitate ER expression in luminal cells in culture[30] showed that CD200$^{low}$-derived differentiated cells are able to express ER (Supplementary Fig. 4c), while CD200$^{high}$ cells fail to expand or express ER under these conditions. This confirms that luminal differentiation is induced in CD200$^{low}$ and CD200$^{high}$ MEPs, while generation of more mature K14$^-$/K19$^+$ luminal cells is confined to CD200$^{low}$ MEP cells.

**MEP cells are founders of luminal epithelial hyperplasia upon *PIK3CA* transformation of CD200$^{low}$ cells.** With the aim of establishing cell lines expressing mutant *PIK3CA*, we selected a biopsy that exhibited strong CD200 staining in terminal ducts and displayed a very clear separation of CD200$^{low}$ and CD200$^{high}$ cells by FACS. Sorted CD200$^{low}$ and CD200$^{high}$ cells from this biopsy were transduced with *hTERT* and shp53 in passage one and two, respectively, followed optionally by *PIK3CA*$^{H1047R}$ in passage three (Supplementary Fig. 5a). Notably, overexpression of *PIK3CA*$^{H1047R}$ without prior knockdown of *TP53* induced cellular senescence, which is neither unforeseen[31], nor compatible with further investigation. While the untransduced parent populations, CD200$^{low}$ and CD200$^{high}$, within few passages became metaplastic and ceased to grow, an extended lifespan was obtained with *hTERT*, and the proliferative capacity was highly increased and to similar extents in CD200$^{low}$ and CD200$^{high}$ cells upon further addition of shp53 irrespective of *PIK3CA* overexpression (Supplementary Fig. 5b). This implies that the introduction of mutant *PIK3CA* does not increase the proliferative capacity compared to cells expressing *hTERT*-shp53 only. That *TP53* was effectively silenced and *PIK3CA* was expressed in the resultant lines was confirmed by RT-qPCR (Supplementary Fig. 5c). These data show that human MEP cells can be successfully immortalized, and to our knowledge, these lines represent the first human normal breast cell lines with an identifiable MEP origin.

To elucidate the effect of mutant *PIK3CA*, we subsequently compared CD200-*hTERT*-shp53 with CD200-*hTERT*-shp53-*PIK3CA*$^{H1047R}$ from CD200$^{low}$ and CD200$^{high}$ origins, respectively. In both lines in ground state conditions, expression of mutant *PIK3CA* resulted in a decrease in K14 and an increase in K19 and α-SMA at both translational and transcriptional levels (Fig. 4a, b). Interestingly, expression of p63, K5, K14, and K17 is retained in all cell lines independent of mutant *PIK3CA* (Supplementary Fig. 6a), and furthermore, CD200-*hTERT*-shp53-*PIK3CA*$^{H1047R}$ cell lines maintain differential expression of *CD200*, *AMIGO2*, *ACTA2*, *FAM126A*, *TXNIP*, *RARRES1*, and *TNFSF10* (Supplementary Fig. 6b). Notably, upon induced differentiation, both CD200$^{low}$-*hTERT*-shp53-*PIK3CA*$^{H1047R}$ and CD200$^{high}$-*hTERT*-shp53-*PIK3CA*$^{H1047R}$ maintain their original phenotypes and only CD200$^{low}$-*hTERT*-shp53-*PIK3CA*$^{H1047R}$ generates mature luminal K14$^-$/K19$^+$ cells (Fig. 4c). The potential of progenitors to further commit to the luminal lineage was revealed by testing the expression of ER, previously shown in mice to be induced as a result of *PIK3CA*$^{H1047R}$ expression in basal cells[32]. Here, *ESR1* is induced by mutant *PIK3CA* in CD200$^{low}$ as well as CD200$^{high}$, but expressed at a significantly higher level in CD200$^{low}$ (Fig. 4c). While the ER-associated marker FOXA1 was expressed in both cell lines, another ER-associated protein, GATA3, was more prominent in CD200$^{low}$ cells compared to CD200$^{high}$ (Supplementary Fig. 6c). This indicates that mutant *PIK3CA* in CD200$^{low}$ supports key progenitor characteristics, such as myodifferentiation, and at the

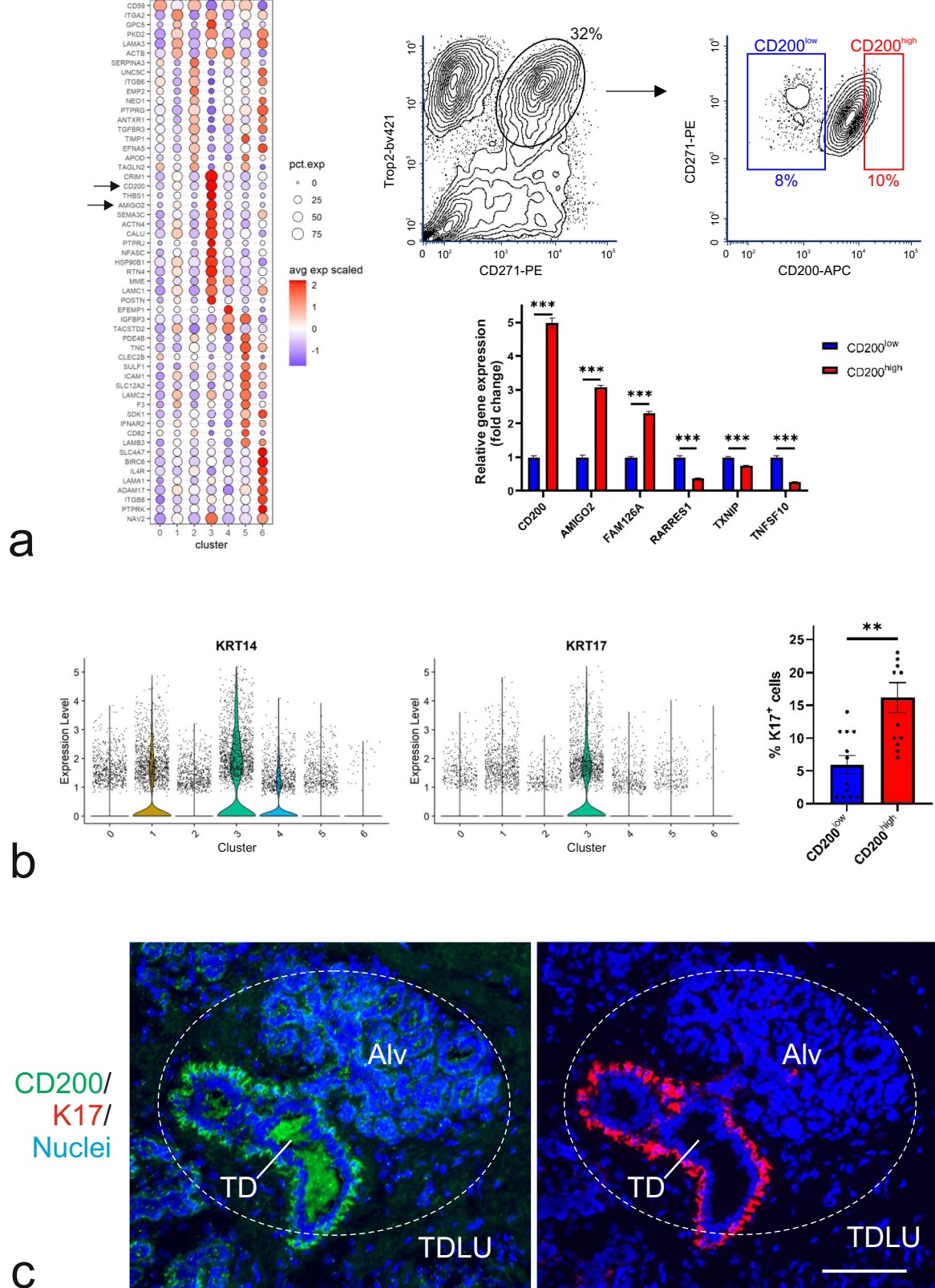

same time pushes the cells towards luminal differentiation, the latter suggesting that human MEP cells in ground state become destabilized upon *PIK3CA*^H1047R expression as in mice[33].

Finally, to address whether *PIK3CA*^H1047R transforms human MEP progenitors, we employed conventional assays for evaluating normal and transformed cell behavior. When tested in soft agar in ground state conditions, mutant *PIK3CA* leads to similar increased frequency and size of colonies from CD200^low and CD200^high cells, respectively (Fig. 5a), thus indicating a transformed phenotype in both. To assess in more detail the level of transformation we employed an organoid assay of human breast morphogenesis[21,34,35]. In this assay, CD200^low as well

**Fig. 2 CD200 is a marker for a distinct MEP subpopulation. a** (left) Cluster-assigned bubble plot of cluster-specific DEGs encoding cell-surface markers confirms expression of *CD200* and *AMIGO2* in cluster 3 (arrows). (right, upper) Contour FACS plots of crude preparation of breast organoids isolated with Trop2 and CD271, followed by gating for MEP cells (encircled with the approximate percentage of cells indicated), subsequently sorted according to CD200 and CD271 to isolate CD200[low] (blue) and CD200[high] (red) cells, respectively. (right, lower) Bar graph of the fold change of normalized relative gene expression by RT-qPCR show upregulation of *CD200*, *AMIGO2* and *FAM126A*, and downregulation of *RARRES1*, *TXNIP* and *TNFSF10* in CD200[high] (red) versus CD200[low] (blue). Bars indicate mean ± SEM (***$p < 0.005$ by multiple *t* tests with Bonferroni correction, $n = 3$). **b** (left) Violin plots of *KRT14* and *KRT17* expression levels reveal upregulation of both in cluster 3. (right) Bar graph of quantification of immunostaining of smears from CD200[low] (blue) and CD200[high] (red) MEP cells confirms a significantly higher frequency (% K17[+] cells) among CD200[high] cells. Bars indicate mean ± SEM (**$p = 0.005$ by two-tailed Mann-Whitney test, $n = 6$ smears from two biopsies). **c** Representative images of cryosection of normal breast stained by immunofluorescence for CD200 (green), K17 (red), and nuclei (blue) show co-localization of CD200 and K17 in the terminal duct (TD) rather than alveoli (Alv) of a TDLU (encircled, scale bar, 100 μm).

CD200[high] transformed MEP progenitors mimicked benign hyperplastic lesions, that is with a basal layer of MEP cells generating multiple layers of luminal cells with a central MUC1 expressing lumen, if present (Fig. 5b and Supplementary Fig. 7). Interestingly, whereas the CD200[low] transformed progenitors generated more keratin K19[−] than K19[+] luminal cells—a previously shown property of normal TDLU-derived progenitors[10]—CD200[high] transformed progenitors generated mostly K19[+] luminal cells (Fig. 5b). This benign behavior of cells in the organoid assay was echoed in a humanized microenvironment xenograft model[10]. As revealed with use of human-specific antibodies to distinguish human-derived structures from the mouse host, in general, CD200[low] as well as CD200[high] cells formed few structures. However, a distinct difference was nevertheless observed: Whereas CD200[high] cells generated very few (only four in total), essentially abortive structures that were all K17[+]/K19[+], CD200[low] generated structures, of which the majority included K17[−]/K19[+] cells (89% K17[−]/K19[+] structures, 8 out of 9 structures, Fig. 6). Upon further scrutiny, the morphology of structures from CD200[low] resembled that of biphasic lesions with an outer layer of K5[+], K17[+], and α-SMA[+] MEP cells surrounding a core of K5[+]/K8/18[+]/K19[+] hyperplastic cells (Fig. 6).

Taken together, these findings suggest that two functionally distinct MEP progenitors exist in human normal breast, including within the terminal ducts of TDLUs, and more importantly, the progenitors differ in susceptibility to transformation. This opens for the hypothesis of a MEP source of breast lesions with mutations shared between the luminal- and MEP compartments, and thus a more important role than hitherto anticipated of the apparently uninvolved layer of MEP cells, which quite often surrounds proliferative luminal lesions (summarized in Fig. 7).

## Discussion

With the purpose of providing an earlier detection of precursor cells and a better prediction of tumor behavior, many efforts over the past decade have been directed towards the identification of cells-of-origin of the different breast cancer subtypes (for review see[36]). While the cell-of-origin of the two major subtypes most likely resides in the luminal compartment, the role of the MEP compartment in tumor evolution remains understudied[37,38]. We here demonstrate by spatial mapping at the single-cell level that the MEP lineage is divided into functionally distinct compartments inherently poised to differentiate into region-specific luminal epithelial cells. The progenitors endure both *hTERT* immortalization and further oncogenic transformation without turning into overtly malignant luminal-like cancer, instead offering a plausible MEP cell of origin of biphasic breast lesions. Although it has been shown that benign lesions like intraductal papillomas have a monoclonal origin[18], answering this question has been severely hampered until now by the lack of human breast MEP cell-based progression series[38]. Our finding offers an

evolutionary explanation to the MEP participation in benign proliferative lesions of known PIK3CA genotype[15], and it parallels the finding in mice of luminal tumor cells in lesions originating in *PIK3CA* destabilized MEP cells[39]. Importantly, however, our findings should not be interpreted in favor of a universal contribution of the MEP lineage to breast cancer as MEP cells repeatedly have been excluded from the malignant clone in the most common tumor types and their immediate precursor lesions[40,41].

Mouse mammary and human breast stem cells reside in the MEP compartment[25,42,43]. These cells are characterized by giving rise to both the luminal and myoepithelial lineage on demand, albeit pending myodifferentiation[10,25,44–46]. However, in general, cell culture fails to support myodifferentiation, and also, MEP differentiation in vivo is sensitive to changes in the microenvironment[47,48]. We previously cultured myodifferentiated MEP cells on mouse 3T3 cell feeders[10]. In the present study, we employed a newly established human breast fibroblast cell line with documented ability to interact with the MEP compartment[49]. In an iterative manner, we further tested a number of factors known to impact stem cell behavior and settled on a supplement of IFNα and RepSox and hypoxic conditions[26–28], here termed Myo[+] conditions. This yielded a biopsy-dependent success rate of up to 70% α-SMA positive MEP cells in the serial passage.

The most obvious difference between mouse and human in terms of the mammary gland and breast anatomy is the presence of prominent TDLUs in the latter. Early autopsy studies and the overwhelming evidence that the majority of breast cancers is K14[−]/K19[+19,50,51] emphasize that any attempt to localize cells-of-origin in the human breast should take into account both anatomy and differentiation repertoire. Here, we sought to improve resolution at the cellular level by use of scRNA-seq of equal numbers of cells sorted from organoids collected under the microscope. By this approach, we compensated for the relative low number of ducts versus TDLUs in most samples of human normal breast tissue[10]. Accordingly, the cells were visualized within the context of TDLUs and ducts, and expression patterns of TDLU- and duct-derived cells were analyzed in regard to the clusters. This in turn was combined with filtration for surface markers and consequently, CD200 localized to the cluster accumulating TDLU-derived cells. Staining of the tissue revealed that within TDLUs, CD200 was often highly expressed in terminal ducts, even if it was low or negative in alveoli. This subtlety offers a plausible explanation to the paradoxical observation at the mRNA level of both CD200 and ductal K17 in the TDLU-enriched cluster 3. As summarized in Fig. 7, CD200[low] cells were found in both K17[−] alveoli and in K17[+] terminal ducts and ducts. Nevertheless, our cell-based assay only gauges for K17[+] cells irrespective of the CD200 status. Whether the K17[−]/CD200[low] cells of alveoli behave the same way as K17[+]/CD200[low] progenitors of terminal ducts or ducts cannot be deduced from the present study. However, in mice alveolar MEP cells are

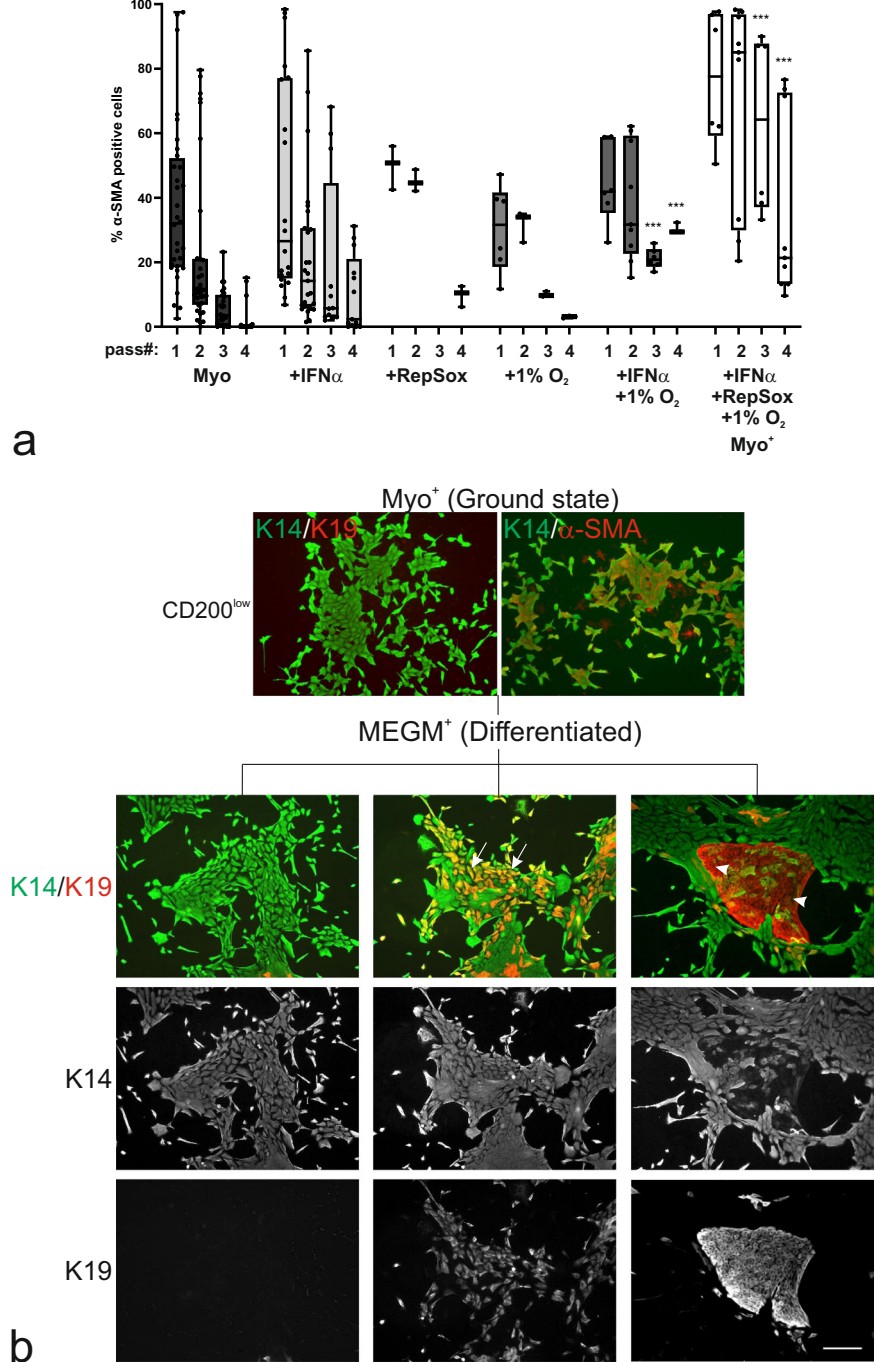

**Fig. 3 Myo⁺ conditions support ground state of MEP progenitors poised for luminal differentiation. a** Whisker boxplot of quantification of the frequency of α-SMA-positive cells (%) in passage 1 to 4 of Trop2⁺/CD271[high] MEP cells on iHBFC[CD105] feeders. In Myo medium (black), Myo medium with IFNα (light grey) or RepSox (medium grey), or in Myo medium under hypoxic conditions (1% O$_2$, medium-dark grey), the frequency of myodifferentiated MEP cells declines with the passage. In Myo medium with IFNα and 1% O$_2$ (dark grey), a significantly higher frequency of myodifferentiated MEP cells is maintained in passages 3 and 4 ($p < 0.005$ by multiple unpaired $t$ tests with Bonferroni correction, compared to Myo medium). In Myo medium with IFNα, RepSox, and 1% O$_2$ (Myo⁺, white) a significantly higher myodifferentiation is obtained in all passages ($p < 0.005$ by multiple unpaired $t$ tests followed by Bonferroni correction, compared to Myo medium), thus defining a reliable ground state condition. Whiskers indicate the minimum and maximum ($n >/= 3$). **b** Micrographs of primary CD200[low] MEP cells stained for K14 (green), K19 (red) and α-SMA (red) show maintenance of K14⁺/K19⁻ and K14⁺/α-SMA⁺ MEP progenitors in Myo⁺ (Ground state). Upon differentiation in MEGM⁺ (Differentiated), the K14⁺/K19⁻ phenotype is maintained (first column), but in addition, colonies comprising K14⁺/K19⁺ (second column, arrows) and K14⁻/K19⁺ cells (third column, arrow heads) appear (scale bar, 100 µm).

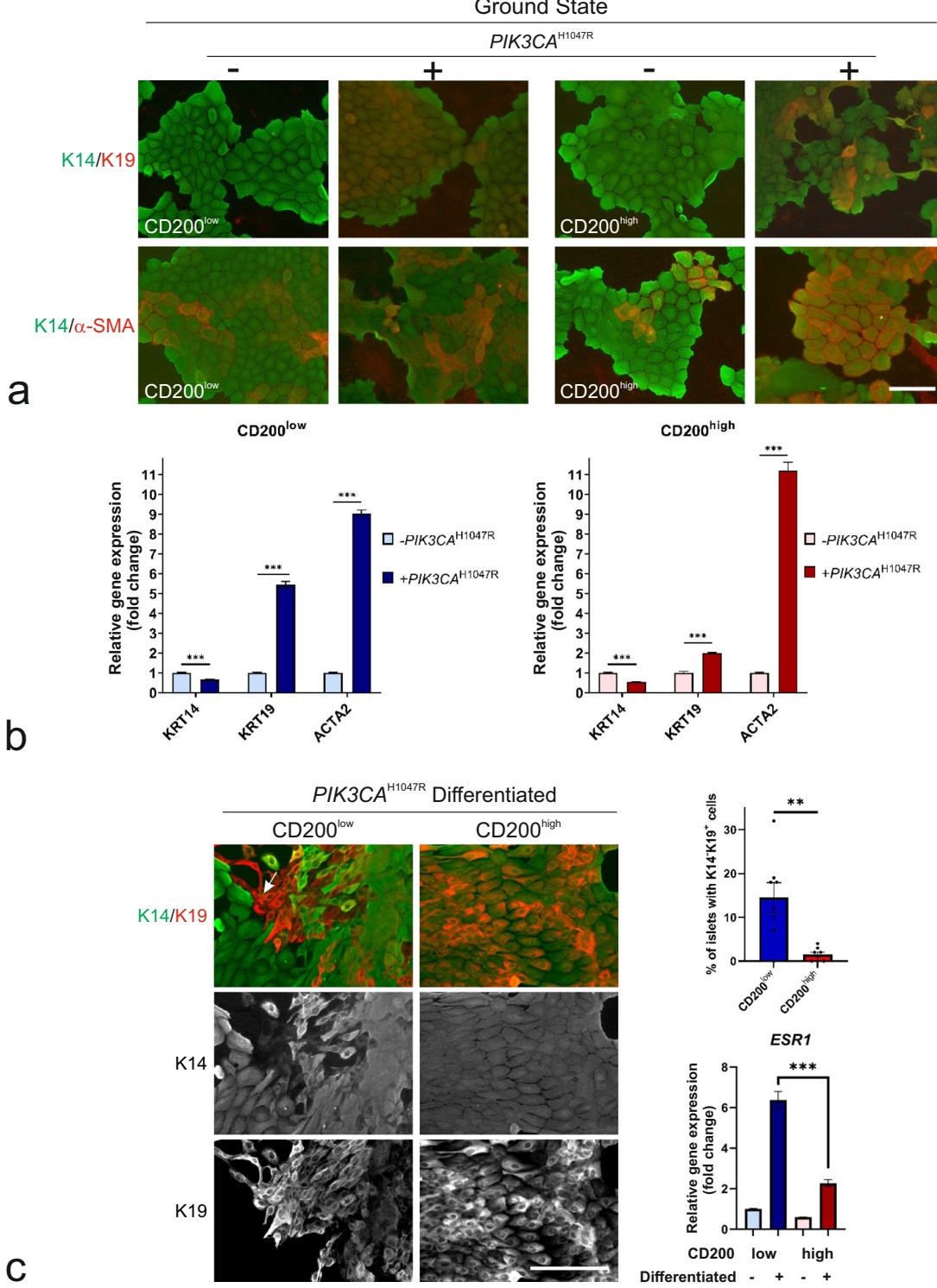

**Fig. 4 Mutant *PIK3CA* destabilizes ground state phenotype without affecting differentiation repertoire. a** Fluorescence micrographs of (left) CD200[low]-*hTERT*-shp53 and (right) CD200[high]-*hTERT*-shp53 without (−) or with (+) mutant *PIK3CA* (*PIK3CA*[H1047R]) in Myo[+] (Ground state) show apparent maintenance of K14 (green) and increase in K14[+]/K19[+] (red) and K14[+]/α-SMA[+] (α-SMA, red) upon *PIK3CA*[H1047R] expression in both cell lines (scale bar, 50 μm). **b** Bar graphs of fold change of normalized relative gene expression by RT-qPCR in CD200[low]-*hTERT*-shp53-*PIK3CA*[H1047R] (blue) and CD200[high]-*hTERT*-shp53-*PIK3CA*[H1047R] (red) strains relative to CD200-*hTERT*-shp53 strains show a somewhat reduced *KRT14* in both lines, increased *KRT19*, and in CD200[low] in particular, a much higher expression of *ACTA2* with mutant *PIK3CA*. Bars indicate mean + SEM (***$p < 0.005$ by multiple unpaired t tests with Bonferroni correction, $n = 3$). **c** (left) Micrographs of fluorescence stainings of (left column) CD200[low]-*hTERT*-shp53-*PIK3CA*[H1047R] and (right column) CD200[high]-*hTERT*-shp53-*PIK3CA*[H1047R] in MEGM[+] (Differentiated) reveal a significantly higher number of colonies with K14[−]/K19[+] (red, arrow) cells in CD200[low] versus CD200[high] (scale bar, 50 μm). (right, upper) Bars indicate mean ± SEM (**$p < 0.01$ by unpaired, two-tailed t test, $n = 8$). (right, lower) Normalized relative gene expression of *ESR1* in CD200[low] (blue) versus CD200[high] (red) by RT-qPCR shows a relative, significant increase of *ESR1* in CD200[low] upon differentiation (+) versus ground state (−). Bars indicate mean ± SEM (***$p < 0.005$ by one-way ANOVA with Tukey's multiple comparison test, $n = 3$).

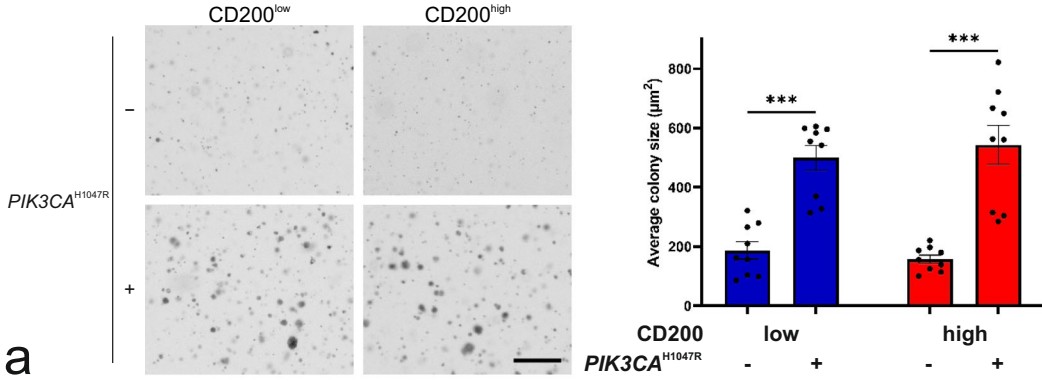

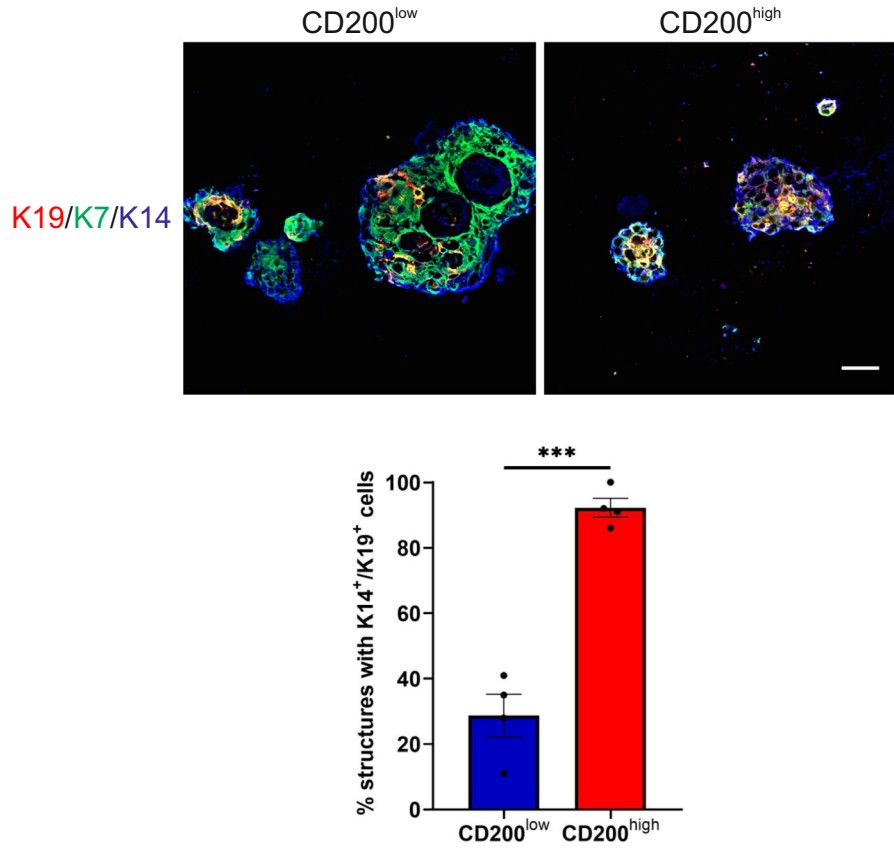

**Fig. 5 Mutant *PIK3CA* partially transforms CD200^low and CD200^high MEP progenitors. a** (left) Phase contrast micrographs of colonies formed in soft agar by (left column) CD200^low-*hTERT*-shp53 (CD200^low) and (right column) CD200^high-*hTERT*-shp53 (CD200^high) without (−) or with (+) mutant PIK3CA (*PIK3CA*^H1047R) in Myo+ at normoxia (scale bar, 500 μm) and (right) quantification of average colony size show that both cell lines form more colonies of larger size with mutant *PIK3CA*. Bars indicate mean ± SEM (***$p < 0.005$ by Kruskal-Wallis test with Dunn's multiple comparison test, $n = 9$). **b** (upper) Micrographs of fluorescence stainings of cryosections stained for K7 (green), K19 (red) and K14 (blue) of structures formed in rBM by CD200^low-*hTERT*-shp53-*PIK3CA*^H1047R (CD200^low) and CD200^high-*hTERT*-shp53- *PIK3CA*^H1047R (CD200^high) (scale bar, 100 μm) and (lower) quantification of structures including K14+/K19+ cells show a significantly higher frequency of structures with destabilized phenotype in CD200^high. Bars indicate mean +/− SEM (***$p < 0.005$ by unpaired, two-tailed *t* test, $n = 4$).

considered as more mature cells having migrated from ductal progenitors[52]. This and other studies promise for an increasing appreciation of the existence of region-specific progenitors in both mouse and human glands[52,53]. Hopefully, the finding of regional MEP markers such as CD200 will lead to more insight into their function in tissue homeostasis and cancer. So far, we know from other tissues that CD200 is implicated in immune

evasion, which is a privilege of somatic stem cells such as those of the bulge of hair follicles[54], and that CD200 is involved in cell-cell communication and TGF-β signaling between neurons and glial cells[55].

*PI3K* is the most frequently mutated gene in human breast cancer on top of *TERT* promoter mutations/amplifications and *TP53* mutations/wtp53 losses, which are considered likely early

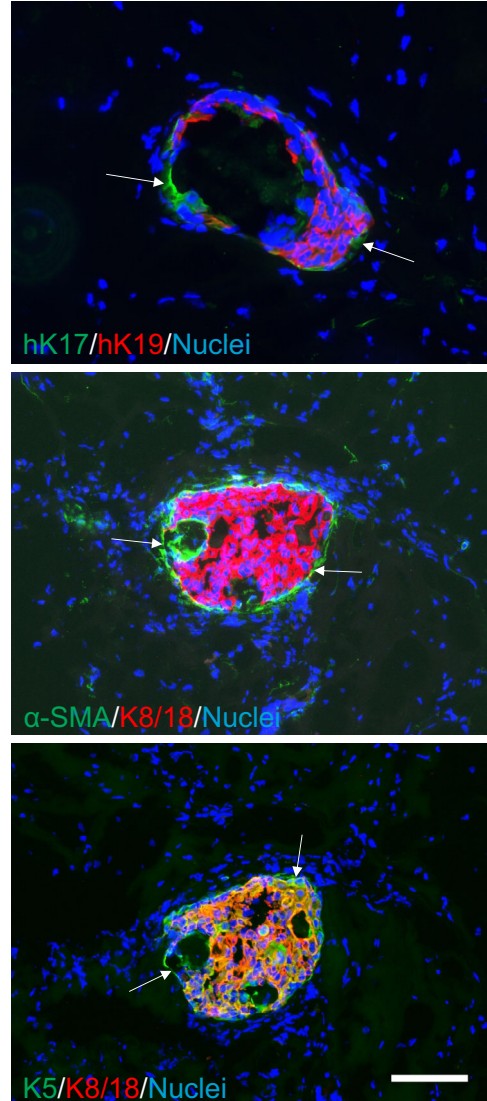

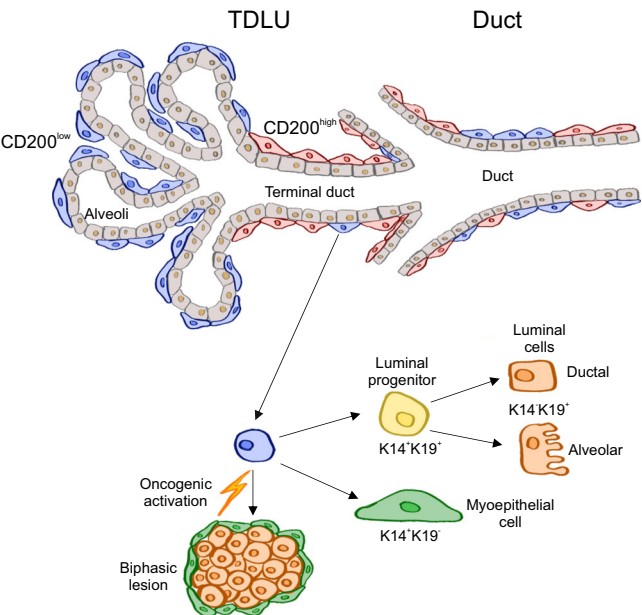

**Fig. 7 Hypothesis for evolution of luminal hyperplastic lesions from CD200$^{low}$ MEP progenitors.** Schematic drawing shows distribution of CD200$^{low}$ MEP cells (blue) in alveoli and CD200$^{high}$ MEP cells (red) in terminal ducts and ducts. CD200$^{low}$ MEP progenitors are multipotent since they can generate K14$^+$/K19$^-$ MEP cells (green), K14$^+$/K19$^+$ luminal progenitor cells (yellow) as well as K14$^-$/K19$^+$ luminal cells of alveoli and ducts (orange). Upon oncogenic activation as shown here by overexpression of *PIK3CA*$^{H1047R}$, CD200$^{low}$ MEP progenitors can give rise to biphasic lesions with luminal hyperplasia.

**Fig. 6 Xenografted *PIK3CA* transformed CD200$^{low}$ MEP progenitors give rise to luminal hyperplastic lesions.** Fluorescence micrographs of structures formed by CD200$^{low}$-*hTERT*-shp53-*PIK3CA*$^{H1047R}$ in vivo in NOG mice stained for human K17, (hK17), K5, α-SMA (green), human K19 (hK19), K8/18 (red), and nuclei (blue) show histology reminiscent of biphasic lesions with an outer layer of K17$^+$, K5$^+$, and α-SMA$^+$ MEP cells (arrows), surrounding a core of hK19$^+$/K5$^+$/K8/18$^+$ hyperplastic cells (scale bar, 100 μm).

events in transformation[56]. We therefore immortalized CD200$^{low}$ and CD200$^{high}$ MEP cells with *hTERT* and shp53 and further transformed the cells with mutant *PI3KCA*. Transformation per se was recorded by anchorage-independent growth in soft agar. We found that transformed MEP cells of both origins had K19 induced reflecting a destabilized phenotype intermediate between MEP and luminal cells. This supports that mutant *PIK3CA* promotes multipotency[57] and illustrates that the effect of mutant *PIK3CA* expression on the MEP lineage is conserved between mouse and human cells. In the mouse mammary gland expression of mutant *PIK3CA* leads to destabilization of lineages, which otherwise under resting conditions are renewed by self-duplication with little or no cross over[32,33]. For obvious reasons, similar experiments cannot be performed in human beings, and until now, cell lines representing normal breast MEP cells, let alone the detail of specific regional origin, have not been

available. Therefore, the most important finding of the present study is that an unequivocally clean and well-defined MEP source upon relevant stimulation gives rise to bona fide, full-blown K14$^-$/K19$^+$ luminal cells if recovered from CD200$^{low}$ MEP cells. K14$^-$/K19$^+$ luminal cells were found in culture and in vivo in nodules reminiscent of biphasic lesions, and it emphasizes the importance of cell-of-origin and ground state phenotype for the differentiation repertoire of destabilized MEP progenitors.

The response to mutant *PIK3CA* reported here for human MEP progenitors compares with that in mice where mutant *PIK3CA* expression in basal cells evokes primarily benign tumors[33]. Importantly, *PIK3CA* in human cells can convey further luminal differentiation comparable to that in mice including ER expression on a MEP background[33]. This represents a long-sought for answer to the question: Can experimental evidence be provided for a role in tumor evolution of the apical most cells in the human breast stem cell hierarchy? We and others have previously approached this question by enrichment or cloning of cells well after immortalization or by use of spontaneously immortal cell lines[2,5,8,38,58–62]. In all instances, however, the MEP phenotype had drifted in culture sufficiently long to facilitate either squamous metaplasia or partial luminal differentiation making lineage tracing impossible. Alternatively, cloning had led to a level of MEP lineage restriction where mutant *PIK3CA* transduction fails to destabilize the cells in the luminal direction[62].

We argue that in order to assess the consequences of the cellular transformation of candidate cells-of-origin the cellular trajectory needs to be perfectly identifiable all the way from tissue heterogeneity to formation of lesions in mouse models. The purpose of the present study was not to generate overtly malignant neoplasias. Clearly, malignant transformation of human cells requires more than mutant *PIK3CA*[3,62]. Instead, we find that the

consequence of targeted transformation of CD200[low] MEP progenitors is biphasic lesions where the aberration is most evident in the luminal compartment in spite of the fact that the notorious founder of the lesion is the MEP lineage. This sheds light on our understanding of lesions where mutant *PIK3CA* involves both the MEP and luminal lineage including benign precursor lesions and overt biphasic tumors such as adenomyoepitheliomas[17,37]. Another perspective of our findings relates to the fact that there is an increasing appreciation of MEP cells as targets for therapy[48]. We believe that the MEP-derived lesions generated in our study lend themselves to screening efforts of pathways responsible for disturbing the intricate reciprocal crosstalk between MEP and luminal epithelial lineages currently believed to precede epithelial neoplasia[48].

## Methods

**Tissue**. Normal breast tissue was obtained from 37 women aged between 13 and 59 years undergoing reduction mammoplasty for cosmetic reasons. Information about donors is restricted to the donor's age at the time of surgery. The tissue was donated with written consent by donors who received information before surgery. The Regional Scientific Ethical Committees (Region Hovedstaden, H-2-2011-052) and the Danish Data Protection Agency (2011-41-6722) reviewed and approved the use and storage of human material. Some of the donated tissue has been included in other studies. Normal breast tissue was cut in smaller pieces for freezing for cryosectioning or dissociated to release epithelial cell clusters, i.e. organoids, as described[63,64]. Organoids were suspended in 90% fetal bovine serum (F7524, Sigma-Aldrich) and 10% dimethyl sulfoxide (D2650, Sigma-Aldrich) and stored in liquid nitrogen.

**FACS**. To isolate MEP cells, organoids were thawed, trypsinized, and filtered as previously described[10]. Single human breast cells were incubated with fluorochrome-conjugated primary antibodies for 45 min at 4 °C. Antibodies included Trop2-bv421 and Trop2-bv510, clone 162-46 (BD Biosciences); CD271-PE, clone Me20.4 (BioLegend); CD271-APC, clone Me40.4 (Cedarlane Laboratories); CD200-APC and CD200-bv421, clone Ox-104 (BioLegend) and AMIGO-AF488, clone S86-36 (Novus Biologicals), all 1:50. After washing and filtration, cells were sorted using a BD FACSAria™ Fusion Cytometer or a BD FACSAria™ III Cytometer equipped with 100 μm nozzles. Regardless of fluorophores, different antibodies recognizing the same antigen provided similar efficiency for sorting.

**scRNA-seq**. scRNA-seq was performed using the 10x Genomics Chromium platform. TDLUs and ducts from three age-matched donors (18 years old, Supplementary Table 2) were picked as described[2], and MEP cells were isolated using FACS as described above. While the parital status of the donors is unknown, biopsies from young donors at the age 18 were selected to reduce the risk of introducing parity-related variability between samples. Subsequently, TDLU-derived and ductal MEP cells were processed using the Chromium Single Cell 3' Reagent Kit v2 or v3. RNA isolation, cDNA amplification, and library preparation were performed according to the manufacturer's instructions (10x Genomics). Sequencing was performed using the Illumina® NextSeq500/550 High Output Kit v2 for 150 cycles according to the manufacturer's instructions and the instructions of the 10x Genomics Chromium Single Cell Kit. Raw files from the sequencing were demultiplexed, aligned to the human reference genome GRCh38-1.2.0.pre-mrna (including introns), filtered, and barcodes and unique molecular identifiers were counted using the 10x Genomics Cell Ranger software. Quality control, filtering, normalization, clustering, and visualization of the data were performed following the "Guided Clustering Tutorial" of the R package Seurat (version 3.0)[22,23]. Cells with unique feature counts over 2,000 or less than 200, and >10% mitochondrial counts were excluded from the analysis. The different datasets were integrated following Seurat's "Integration and Label Transfer" vignette[23]. Briefly, cells with similar gene expression patterns across the three datasets were identified, and based on this, the datasets were corrected for technical discrepancies. Subsequently, analysis and clustering were performed on the integrated and corrected dataset[23]. DEGs between the clusters were identified when a gene was expressed in at least 20% of the cells of one cluster (min.pct = 0.2) and the average fold change was >0.25 (thresh.test = 0.25). DEGs are based on Wilcoxon rank-sum tests followed by Bonferroni correction for multiple testing and only significant genes ($p < 0.05$) were classified as DEGs. To compare clusters to previously published datasets, DEGs from each cluster were correlated with annotated gene sets using the "Investigate Gene Sets" function of the Molecular Signatures Database (https://www.gsea-msigdb.org/gsea/msigdb/).

**Cell culture**. Freshly sorted primary human breast MEP cells and MEP cell lines (see below) were cultured on immortalized CD105[high] human breast fibroblasts (iHBFC[CD105])[49] at 37 °C in a humidified atmosphere with 5% $CO_2$ in modified breastoid base medium without HEPES (BBM)[65], composed of DMEM/F12

medium (Dulbecco´s Modified Eagle´s Medium/Nutrient Mixture F-12, 1:1, Gibco, 21041025) with 2 mM glutamine, 50 μg ml[−1] gentamycin, 1 μg ml[−1] hydrocortisone, 9 μg ml[−1] insulin, 5 μg ml[−1] transferrin, 100 μM ethanolamine, 20 ng ml[−1] basic fibroblast growth factor, 5 nM amphiregulin with the addition of 10 μM Y27632, 180 μM adenine, 20 μl ml[−1] serum replacement B27 (Myo medium[66], but without Na-Selenite), with further addition of 10 ng ml[−1] IFNα2 A (H6041, Sigma-Aldrich), and 25 μM RepSox (R0158, Sigma-Aldrich) in hypoxic conditions, 1% $O_2$, here termed Myo[+]. Cell lines were primed for differentiation (see below) by the addition of 10% FCS to the medium. MEP cells were passaged every 5–7 days by brief trypsination to remove fibroblast feeders followed by the release of MEP cells, which were counted manually and seeded at a density of 4000 cells cm[−2] on new feeders. iHBFC[CD105] were cultured as described[49] and used when 60–80% confluent between passage 10 and 50.

**Differentiation assay**. Differentiation capacity was assessed by switching either freshly sorted cells or cells cultured on iHBFC[CD105] feeders to collagen-coated flasks in Mammary Epithelial Cell Growth Medium (MEGM™, CC-3151, and CC-4136, Lonza,) supplemented with 20 ng ml[−1] epidermal growth factor, 4 μg ml[−1] heparin, 20 ng ml[−1] basic fibroblast growth factor, 20 μl ml[−1] B27, and 500 nM A83-01 at 37 °C and 5% $CO_2$[29], here termed MEGM[+]. After 10 days, cells were stained as described below for MEP markers K14 and K17 and luminal marker K19. The frequency of MEP, luminal and double-positive islets were evaluated using a Leica DM5500B fluorescence microscope, and quantified by manually counting islets containing K19[+]/K14[−] cells, islets containing K14[+]/K19[+] double-positive cells, and islets completely negative for K19 (K14[+]/K19[−]). To assess the expression of ER, differentiated cells were cultured in TGFβR2i-1 medium[30].

**Lentiviral production and transduction**. To immortalize and transform human MEP cells, lentiviral vectors of pLV-hTERT-IRES-Hygro (a gift from Tobias Meyer, Addgene plasmid #85140)[67], pLVUH-shp53-eGFP (a gift from Patrick Aebischer & Didier Trono, Addgene plasmid #11653)[68], and pHAGE-PIK3CA-H1047R-puro (a gift from Gordon Mills & Kenneth Scott, Addgene plasmid #116500)[69] were used for gene transduction into CD200[low] and CD200[high] MEP cells. Lentiviral particles were produced by co-transfection of HEK293T cells with packaging vectors containing vesicular stomatitis virus glycoprotein (pCMV-VSVG) and pCMV-ΔR8.9 (gifts from Frederik Vilhardt, University of Copenhagen) and a lentiviral vector encoding the gene of interests using the calcium phosphate method. Culture supernatants containing lentivirus were collected 48 h post-transduction and filtered through 0.45 μm low protein binding PVDF membrane (Merck). In order to produce high-titer stocks, supernatants were concentrated by centrifugation using vivaspin ultrafiltration spin columns (Sartorius). Targeted cells were transduced with viral supernatants (multiplicity of infection < 0.3) for 3 days. Successfully transduced cell populations were obtained by selection with applicable antibiotics (hygromycin, 67 μg ml[−1] and puromycin, 0.67 μg ml[−1]) or FACS based on eGFP. Freshly isolated CD200[low] and CD200[high] MEP cells seeded in Myo[+] medium on human fibroblast feeders were grown to 30% confluency before transduction with lentiviral particles of *hTERT*. Upon selection with hygromycin, *hTERT* expressing cells were transduced with either lentiviral shp53 and subsequently with *PIK3CA[H1047R]*.

**Immunocytochemistry and immunohistochemistry**. Cultured cells were washed with phosphate-buffered saline with calcium and magnesium (PBS[+]) and fixed for 5 min in 3.7% formaldehyde, washed with PBS, fixed for 5 min with methanol/acetone at −20 °C, washed, and permeabilized twice for 7 min with 0.1% Triton-X100. Cell smears and cryostat section were fixed for 10 min with 3.7% formaldehyde, washed with PBS, and permeabilized with 0.1% Triton X-100. Blocking was performed using 10% goat serum in PBS for at least 5 min. Cells/sections were incubated for 1 h with the primary antibody in 10% goat serum in PBS, washed three times, and incubated for 30 min with secondary antibody in 10% goat serum in PBS. For peroxidase staining, cells were incubated for 30 min with Ultravision ONE HPR polymer (TL-125-PHJ, Thermo Scientific) instead of secondary antibody. Cells were then washed with PBS and incubated for 10 min with 3'3-diaminobenzidine and 1:1000 30% hydrogen peroxide. The overview of antibodies used is summarized in Supplementary Table 3. Nuclei were counterstained with DAPI (P36935, Invitrogen).

**RNA isolation and RT-qPCR**. RNA isolation and RT-qPCR were performed as described previously[49]. Briefly, RNA was isolated using the Direct-zol™ RNA Micro or MiniPrep kits (Zymo Research) according to the manufacturer's instructions. Reverse transcription was performed using the High Capacity RNA-to-cDNA kit (Applied Biosystems). For RT-qPCR, the TaqMan™ gene expression system (Applied Biosystems) was used. The genomic means of expression of housekeeping genes *GAPDH*, *TBP*, *ACTB*, *PGK1*, *HPRT1*, and *TFRC* were used for normalization purposes. Data was analyzed using the software Bio-Rad CFX Maestro 1.0 (Bio-Rad, version 4.0). TaqMan probes included *TNFSF10*, Hs00921974_m1; *KRT14*, Hs00265033_m1; *TP63*, Hs00978343_m1; *EPCAM*, Hs00901885_m1; *ELF5*, Hs01063022_m1; *KRT19*, Hs00761767_s1; *ACTA2*, Hs00909449_m1; *ESR1*, Hs00174860_m1; *TP53*, Hs01034249_m1; *PIK3CA*, Hs00907957_m1; *GAPDH*,

Hs02758991_g1; *TBP*, Hs00427621_m1; *ACTB*, Hs01060665_g1; *PGK1*, Hs00943178_m1; *HPRT1*, Hs99999909_m1 and *TFRC*, Hs00951083_m1.

**Soft agar assay**. Anchorage-independent growth of transformed MEP cell lines was assessed as described previously[70]. Briefly, a base layer of 0.5% low gelling agarose solution (A9045, Sigma-Aldrich) was solidified in each well of a 6-well-plate. 100,000 cells were resuspended in a 0.35% agarose solution and seeded on top of the base layer. After solidification of the top agar layer, Myo+ medium was added and cells were incubated for three weeks at normoxia at 37 °C and 5% $CO_2$. Gels were stained using a 0.4% crystal violet solution in 50% ethanol and imaged at low magnification using a Zeiss Axio Zoom.V16 brightfield microscope. Colony numbers and sizes were calculated using the Intellesis function of the Zeiss ZEN software (blue edition, version 3.2).

**Recombinant basement membrane (rBM) cultures**. Three-dimensional cultures of CD200 MEP cell lines were established by resuspending 10,000 cells in 100 μl Matrigel (356231, Corning). The cell-Matrigel-suspension was carefully seeded in preheated 24-well plates and the medium was added after solidification of the gels. rBM cultures were maintained in MEGM+ medium at 37 °C and 5% $CO_2$ and the medium were changed twice weekly. After 2–3 weeks, cultures were snap-frozen in n-hexane for cryo-sectioning.

**Mouse xenografts**. All experiments involving animals have been approved by the Danish Animal Experiments Inspectorate with reference to 2017-15-0201-01315. To assess the capability of cells to grow in vivo, $1 \times 10^6$ MEP cells were mixed with $2.5 \times 10^5$ human breast fibroblasts and suspended in a 1:1 mixture of collagen and Matrigel as described[10]. Cell suspensions were injected subcutaneously in three 6–10 week old female NOD.Cg-*Prkdc*$^{SCID}$*Il2rg*$^{tm1sug}$ (NOG) mice (Taconic) per transformed MEP cell line in the area of the abdominal fourth mammary gland. Drinking water was supplemented with 0.67 μg ml$^{-1}$ 17β-estradiol. After two months, mice were sacrificed, mammary glands were excised and snap frozen in n-hexane for cryo-sectioning.

**Statistics and reproducibility**. All statistical analysis were performed using the softwares RStudio (version 1.2.5001 and version 3.6.2) or GraphPad Prism (version 9.0.0). Data were tested for normal distribution using Shapiro-Wilk and Kolmogorov-Smirnov tests. Tests to determine significant differences between datasets were chosen separately for each experiment and are specified in the figure legends. Significance is indicated as follows: $p$ values >= 0.05, not significant (ns); $p < 0.05$, *$p < 0.01$, **$p < 0.005$, ***.

**Reporting summary**. Further information on research design is available in the Nature Research Reporting Summary linked to this article.

## Data availability

The scRNA-seq raw data are available at the European Genome-Phenome Archive (EGA) with accession ID EGAS00001005933. Source data for all graphs and charts are available in Supplementary Data 3.

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

## Acknowledgements

We gratefully acknowledge the expert technical assistance from Tove Marianne Lund, Lena Kristensen, and Anita Sharma Friismose. Capio CFR Hospitaler (Benedikte Thuesen and Trine Foged Henriksen) and the donors are acknowledged for providing breast biopsy material. The Core Facility for Integrated Microscopy (University of Copenhagen) is acknowledged for confocal microscope accessibility. We thank Gelo Dela Cruz and the DanStem Flow Cytometry Platform for access to FCS Express to analyze flow cytometry data. Furthermore, we thank Helen Neil and the DanStem Genomics Platform for technical expertise, support, and the use of instruments. Finally, we acknowledge Konstantin Khodosevich, Ulrich Pfisterer, and Andrea Asenjo Martinez for access to and support with scRNA-seq equipment and Samuel Demharter and Katharina Theresa Kohler for assistance with bioinformatics analyses. This work was supported by Novo Nordisk Fonden (NNF17CC0027852) and Danish Research Council grant 10-092798 (to DanStem), Familien Erichsens Mindefond and Vera og Carl Johan Michaelsens Legat (to J.K.), Toyota-Fonden Denmark and Anita og Tage Therkelsens Fond (to R.V.), Harboefonden, Else og Mogens Wedell-Wedellborgs Fond, Danish Cancer Society Grant R146-A9257 and Dagmar Marshalls Fond (to L.R.-J.), Novo Nordisk Fonden (NNF18CC0033666 (to N.G.), and the Kirsten and Freddy Johansens Fond (to O.W.P).

## Author contributions

N.G., J.K., L.R.-J., and O.W.P. designed research; N.G., J.K., R.V., N.G., and O.W.P. performed research; N.G., J.K., L.R.-J., and O.W.P. analyzed data, N.G., L.R.-J., and O.W.P wrote the paper. All authors read, revised, and approved the final version of the manuscript.

## Competing interests

The authors declare no competing interests.
