## [Peer Review File · Communications Biology]

Reviewers' comments:

Reviewer #1 (Remarks to the Author):

Goldhammer et al describe characteristics of myoepithelial progenitors (MEPs) of the breast that are present in ductal unit and terminal duct lobular units. Authors purified distinct population of cells from reduction mammoplasty samples and then characterized purified cells using single cell sequencing. They describe distinct properties of CD200-low and CD200-high and suggest that CD200-low cells are multipotent, whereas CD200-high cells are bipotent.

Authors have done extensive analysis using primary tissues and for most part, data support conclusions. However, there are major concerns in few figures, which are most likely artifacts of growth conditions.

1) Authors claim that CD200-low cells expressing mutant PIK3CA can differentiate into K14-/K19+ cells under 2D layer feeder layer differentiation condition (Fig. 4C). However, under organoid growth condition in Figure 5B, it is CD200-high cells that differentiate into K14-/K19+ cells at a higher rate. With these discrepancies, it is hard to conclude that CD200-low cells are multi-potent, whereas CD200-high cells are bipotent. Furthermore, in vivo relevance of the findings are questionable if differentiation properties of these two populations of cells are dependent on growth conditions.

2) Double immunofluorescence with ER/K19, GATA3/K19 and FOXA1/K19 is needed to further document differentiation into hormone responsive mature luminal cells.

3) PIK3CA mutant overexpression studies need to be interpreted with caution. None of the studies involved only hTERT+ mutant PIK3CA. All transfection studies involved knockdown of p53. Since p53 is mutated at lower frequency in breast cancers with PIK3CA mutation, all observations are a result of interplay between PIK3CA activation and lack of p53 instead of the property of mutant PIK3CA alone.

4) Results presented in supplementary Fig. 6 are obtained with CD200-low and CD200-high cells that differ in CD200 expression by only 15%. Therefore, data are not robust.

Reviewer #2 (Remarks to the Author):

With detailed cell sorting, single cell RNA sequencing, and cell culture experiments, the authors characterize expression programs in breast myoepithelial cells, and discover/utilize CD200 expression as a differentiating marker. They have built a novel in vitro cell culture model maintaining longer myoepithelial differentiation. Differences in pluripotency and the ability to recapitulate multi-layered hyperplasia like structures upon transfection are studied and described.

It was not clear from the Results text that most experiments were performed on human breast cells from mammoplasties that were first grown as organoids. This should be clarified throughout.

The authors produce and characterize immortalized cell lines using hTERT, p53 silencing and mutant PIK3CA (H1047R) and provide comparisons of CD200low and CD200high progenitors. They also use control line without the 3rd step of PIK3CA mutation. A better control would be a cell line with a 3rd step of transfection by PIK3CA wild type (eg would the inevitably higher transfected expression levels of PIK3CA wild type have similar effects?). Images from the experiments +/- PIK3CA mutation seem to be at different cell density, which can influence differentiation and expression programs.

Existing data on PIK3CA mutations in myoepithelial vs luminal populations, if any, should be discussed. Although a study of papillary lesions, the paper of Mishima PMID: 29454754 could provide a starting point, or other sorting/single cell sequencing experiments. Such data would substantiate the immortalization model.

The number of abbreviations makes the manuscript hard to read, but I defer to editorial policy (TD, DEG, etc).

Reviewer #3 (Remarks to the Author):

In this study, Goldhammer and collaborators aimed to further elucidate the role of mammary progenitor cells. The authors used an elegant combination of cell extraction from the terminal ductular units, scRNA-seq, and imaging to identify a subpopulation of cells with differentiation ability.

In all these years there is an ongoing debate trying to identify “truly” progenitor cells in the breast; therefore any study which works in that direction is interesting, as it can provide further evidence for the presence of a mammary hierarchy.

I only have minor comments for this paper.

- Line 55: the authors describe their previous paper, where TDLU-derived MEPs can give rise to K19+ or K19- cells. However, they fail to mention this is age-dependant. This should be included to avoid giving the impression of a generic behaviour of the cells during the totality of the life of the individual.

- Samples are taken from reduction mammoplasty, which is often skewed by different BMIs. Are the BMI and the age of these women of significant difference? High BMI has been shown to have a protective effect on younger women, and a detrimental effect on older individuals. I understand the authors may not have this information, but it would be good practice to include it when analysing reduction mammoplasty.

However, considering the difference in heterogeneity which is seen in the TDLU according to different ages (as per their previous publication), at least the age, if not the BMI, should be reported here, and differences in ages on the results (if any difference is present) should be described and discussed.

- Related to this, for scRNA sequencing, samples from 18 year old patients were chosen, but it is not explained why this particular age was chosen.

- Smooth muscle actin alpha is canonically abbreviated to either ACTA2 or SMA. I would suggest replacing either of these with sm actin, which is a very unusual abbreviation. Also, in most of the text, the protein is abbreviated, but then occasionally, such as in line 138, the full name appears again.

- I am not an expert in cluster analysis, so I apologise in advance for this comment, if not relevant. It is not clear to me the difference between Suppl Table 2 and Suppl Table 3. By reading the table legend, I thought the latter was only a subset of the first, but I have noticed that the data is different between the two tables, for the same genes within the same clusters. Could you please improve the figure legend so it is clear the difference between the two set of data, or how the data is derived from the previous ones? Is the further division into anatomic regions the reason for the difference?

PRDX1 1.54E-231 0.438438467 0.939 0.819 3.68E-227 0

PRDX1 6.93E-222 0.429187968 0.939 0.823 1.65E-217 0

- Line 105: The authors screened several antibodies and determined that CD200 was “superior” to AMIGO2. How did they defined “superiority”? Which outcomes were they looking for after IHC or FACS (I guess specificity, or intensity?). The authors would need to explain this in the methods, and possibly include a supplementary figure with at least a couple of representative images to allows us to understand their choice of antibody/protein.

- Line 108: depending on the biopsy, the percentage of CD200low cells accounts to a different percentage of MEP. How would the authors explain this difference?

- Line 119: Could the authors explain how was immunostaining (Fig 1C) quantified in the

methods? Also, the choice of violin plot in Fig 1B is good, but the overlay with all the data points makes it hard to read. I would suggest the authors to remove the individual dots and just include the n numbers near the x axis.

Finally, was immunostaining also tested for K14? If not, the text in line 120 should be edited to: "This was confirmed for K17 by immunostaining".

I would also recommend to use the same nomenclature in the figure, so either K17 or KRT17. Maybe the use of K17/KRT17 could be adopted, if the authors want to use two terms.

- Line 123: "came out strongest" does not sound technical. Please edit to "the intensity of the signal of CD200 was strongest", or "the expression of CD200, as determined by immunostaining, was higher".

- Suppl Fig 2B. The bottom scheme is not very clear. I understand the comparison, but it is not clear what does the length of the line represent. Are those different biopsies? I think it is quite confusing, and should be either improved for clarity or visualised in a different way.

- Line 142: please state how the expression was measured here (immunohistochemistry?)

In relation to this, Fig 3A should include statistical analysis to determine the significant difference in the expression of actin in the last group compared to the others. Comparing the slope of decrease during passages could be a good way to measure this. Also, the variation between samples is so big that I would recommend using a boxplot rather than a bar chart, as it would be more informative.

Also, how was the induction of hypoxia tested after incubation with lower oxygen? Did the authors test the expression of markers such as HIF1a? This should be added in the method section.

- Line 174: The authors have transduced the cells with 3 constructs, subsequently: hTERT, shp53 and PIK3CA. They then observed elevated proliferation in both hTERT/shp53 and hTERT/shp53/PIK3CA - transduced cells. From the text, it does not look like the authors have cells with only hTERT/PIK3CA. I therefore wonder how they can explain their statement saying: "... the introduction of mutant PIK3CA does not increase the proliferative capacity per se". Assuming that "per se" indicates without either hTERT, shp53, or both, proliferation data of cells transduced with PIK3CA or hTERT/PIK3CA should be shown before that statement is made.

- Line 187: "remain faithful to their origins" - please edit with a more appropriate technical language

- Line 266: remove the word keratin, since the abbreviation is present

- Line 413: Invitrogen. Spelling mistake.

Answer to reviewer comments:

We thank the referees for their time and effort in reading our manuscript and for providing constructive feedback and helpful suggestions. Below, please find a point-by-point response to the criticism raised.

Comment	Response
Reviewer 1	
Goldhammer et al describe characteristics of myoepithelial progenitors (MEPs) of the breast that are present in ductal unit and terminal duct lobular units. Authors purified distinct population of cells from reduction mammoplasty samples and then characterized purified cells using single cell sequencing. They describe distinct properties of CD200-low and CD200-high and suggest that CD200-low cells are multipotent, whereas CD200-high cells are bipotent. Authors have done extensive analysis using primary tissues and for most part, data support conclusions. However, there are major concerns in few figures, which are most likely artifacts of growth conditions.	We thank the reviewer for making a thorough analysis of the manuscript. We believe that the reviewer's concerns about few of the figures are due in part to an unfortunate mistake of our own and in part to inclusion of less representative images. In the revised version of the manuscript, the mistake has now been corrected and figures have been revised to include images that are more representative. Please see answers to the specific points below.
1) Authors claim that CD200-low cells expressing mutant PIK3CA can differentiate into K14-/K19+ cells under 2D layer feeder layer differentiation condition (Fig. 4C). However, under organoid growth condition in Figure 5B, it is CD200-high cells that differentiate into K14-/K19+ cells at a higher rate. With these discrepancies, it is hard to conclude that CD200-low cells are multipotent, whereas CD200-high cells are bipotent. Furthermore, in vivo relevance of the findings are questionable if differentiation properties of these two populations of cells are dependent on growth conditions.	We are grateful to the reviewer for raising this point. Unfortunately, we made an error labelling the Y-axis in Figure 5B. As stated in the figure legend in the original version of the manuscript (p. 29, l. 692), the label of the Y-axis should read “% structures with K14⁺/K19⁺ cells”. We found that a lower frequency of structures formed by CD200^{low} in 3D are double positive, and that the CD200^{low} cells readily express K19. As such, CD200^{low} cells exhibit a broader differentiation repertoire. We apologize for the confusion and we have corrected the Y-axis in Figure 5B in the revised version of the manuscript.

2) Double immunofluorescence with ER/K19, GATA3/K19 and FOXA1/K19 is needed to further document differentiation into hormone responsive mature luminal cells.

This is an excellent suggestion.

We have performed additional experiments to investigate the expression of ER, FOXA1, and GATA3 in differentiated cultures of CD200^{low} and CD200^{high} MEPS transduced with *hTERT-shp53-PIK3CA^{H1047R}*. We used peroxidase staining rather than fluorescence stainings due to higher sensitivity of the former method. We could, however, not detect ER protein in cultures of differentiated CD200-*hTERT-shp53-PIK3CA^{H1047R}*. Instead, we found that FOXA1 was widely expressed in differentiated cultures. We also observed occasional expression of GATA3, which was more strongly expressed in CD200^{low} cells compared to CD200^{high}. To illustrate this, we have added images of peroxidase stainings of FOXA1 and GATA3 in Supplementary Fig. 6C, and the text has been changes accordingly at p. 10, l. 193-195:

“While the ER-associated marker FOXA1 was expressed in both cell lines, another ER-associated protein, GATA3, was more prominent in CD200^{low} cells compared to CD200^{high} (Supplementary Fig. 6C).”

	In general, ER expression is difficult to achieve in culture. Therefore, we transferred differentiated CD200^{low} and CD200^{high} myoepithelial cells to conditions that we have previously described to maintain ER expression in luminal cells in culture (Hopkinson et al., Oncotarget 2017). Whereas CD200^{high} cell did not expand under these conditions, CD200^{low} cells grew well and in addition, occasionally expressed ER. This point has now been added to the text and illustrated in an additional Supplementary Fig. 4C. The text at p. 8/9, l. 160-163 now reads: “Finally, culture of differentiated CD200^{low} MEPs in a medium described to facilitate ER expression in luminal cells in culture³⁰ showed that CD200^{low}-derived differentiated cells are able to express ER (Supplementary Fig. 4C), while CD200^{high} cells fail to expand or express ER under these conditions.” 3) PIK3CA mutant overexpression studies need to be interpreted with caution. None of the studies involved only hTERT+ mutant PIK3CA. All transfection studies involved knockdown of p53. Since p53 is mutated at lower frequency in breast cancers with PIK3CA mutation, all observations are a result of interplay between PIK3CA activation and lack of p53 instead of the property of mutant PIK3CA alone.	We agree with the reviewer’s comment. We did try to overexpress PIK3CA^{H1047R} on an hTERT mutant background without concurrent knockdown of p53. However, although cells were successfully transduced and survived selection, hTERT-PIK3CA^{H1047R} cells ceased to grow and senesced. Such PIK3CA^{H1047R} oncogene-induced senescence has previously been reported by others (Chakrabarty et al. Carcinogenesis 2019). We added this observation on p. 9, l. 171-172: “Notably, overexpression of PIK3CA^{H1047R} without prior knockdown of TP53 induced cellular senescence, which is neither unforeseen³¹ nor compatible with further investigation.” Since a benign phenotype was observed only in cells expressing hTERT-shp53-PIK3CA^{H1047R}, and not in cells overexpressing hTERT-shp53 alone (Figure 5), we believe that PIK3CA^{H1047R} is indeed the main driver of the observed phenotype.
4) Results presented in supplementary Fig. 6 are obtained with CD200-low and CD200-high cells that differ in CD200 expression by only 15%. Therefore, data are not robust.	We do believe that the data are robust, as we have performed many more experiments than those included in the manuscript. In the original version of the manuscript, Supplementary Fig. 6B represented three

technical replicates. To strengthen the data further, we now have replaced the figure with a graph representing four biological replicates of cells harvested at three different passages (P6, P11, P13, P23), and the figure legend has been changed accordingly. As indicated, the differences in gene expression between CD200^{low} and CD200^{high} cell lines are statistically significant irrespective of passage, thus confirming that these data are indeed robust.

Reviewer 2

With detailed cell sorting, single cell RNA sequencing, and cell culture experiments, the authors characterize expression programs in breast myoepithelial cells, and discover/utilize CD200 expression as a differentiating marker. They have built a novel in vitro cell culture model maintaining longer myoepithelial differentiation. Differences in pluripotency and the ability to recapitulate multi-layered hyperplasia like structures upon transfection are studied and described.

It was not clear from the Results text that most experiments were performed on human breast cells from mammaplasties that were first grown as organoids. This should be clarified throughout.

There is nothing to clarify, since human breast cells from reduction mammaplasties were not cultured as organoids prior to experiments. "Organoids" describe the small fragments of breast tissue that result from collagenase digestion of reduction mammaplasty tissue as illustrated in Figure 1A and described in Methods. To avoid confusion we have modified the text in the Methods section to say "...release epithelial cell clusters, i.e. organoids, as described..." (p. 16, l. 331).

The authors produce and characterize immortalized cell lines using hTERT, p53 silencing and mutant PIK3CA (H1047R) and provide comparisons of CD200^{low} and CD200^{high} progenitors. They also use control line without the 3rd step of PIK3CA mutation. A better control would be a cell line with a 3rd step of transfection by PIK3CA wild type (eg would the

We thank the reviewer for this suggestion. The main aim of PIK3CA^{H1047R} overexpression here was to transform distinct populations of myoepithelial cells and compare resulting lesions. It has been shown previously that transformation of normal human breast epithelial cells is difficult to achieve and commonly requires overexpression or knockdown of several oncogenes or tumor suppressors, respectively (Keller et al. PNAS 2012;

inevitably higher transfected expression levels of PIK3CA wild type have similar effects?).	Dekkers et al. J Natl Cancer Inst 2019). Studies conducted in mice showed that PIK3CA^{wt} overexpression is not sufficient to transform murine mammary epithelial cells (Koren et al. Nature 2015), which led us to reason that PIK3CA^{wt} overexpression would similarly not be sufficient to reach our goal of transforming normal human breast myoepithelial cells. Since in the present study we focused on comparing CD200^{low} and CD200^{high} myoepithelial cells rather than evaluating the transformative power of PIK3CA, we believe that the presented controls are appropriate to substantiate our data.
Images from the experiments +/- PIK3CA mutation seem to be at different cell density, which can influence differentiation and expression programs.	We agree that cell density can have a major effect on differentiation and expression programs. All experiments were performed at similar cell densities (60-70% confluent for stainings, 70-80% confluent for RNA isolation and splitting of cells). We realize, however, that the regions acquired for images in Figure 4A did not display this adequately. In the revised version of the manuscript, we have therefore replaced the images with images of CD200^{low}-hTERT-shp53-PIK3CA^{H1047R} and CD200^{high}-hTERT-shp53-PIK3CA^{H1047R} acquired from the same cultures, albeit from fields that are directly comparable in terms of cell density.  The figure displays a 2x4 grid of fluorescence microscopy images. The top row shows K14/K19 staining (green) in CD200^{low} cells, and the bottom row shows K14/α-SMA staining (green) in CD200^{low} cells. The columns are organized as follows: the first two columns are labeled 'Ground State' and the last two are labeled 'PIK3CA^{H1047R}'. Within each of these two groups, the first column is marked with a '-' sign and the second with a '+' sign, representing different cell density conditions. The images show that PIK3CA^{H1047R} expression leads to increased α-SMA staining in CD200^{low} cells, and this effect is more pronounced at higher cell density (+).
Existing data on PIK3CA mutations in myoepithelial vs luminal populations, if any, should be discussed. Although a study of papillary lesions, the paper of Mishima PMID: 29454754 could provide a starting point, or other sorting/single cell sequencing experiments. Such data would substantiate the immortalization model.	That is an excellent suggestion. We touched upon this topic in the introduction and included the publication by Mishima et al. (p. 4, l. 59). Furthermore, the following sentence was added in the discussion on p. 12, l. 238-239: “Although it has been shown that benign lesions like intraductal papillomas have a monoclonal origin¹⁸, answering this question has been severely hampered until now by the lack of human breast MEP cell-based progression series³⁸.”
The number of abbreviations makes the manuscript hard to read, but I defer to editorial policy (TD, DEG, etc).	We excluded the abbreviation “TD” from the manuscript and exchanged “sm actin” (smooth muscle actin) with the more commonly used “SMA”.

Reviewer 3	
In this study, Goldhammer and collaborators aimed to further elucidate the role of mammary progenitor cells. The authors used an elegant combination of cell extraction from the terminal ductular units, scRNA-seq, and imaging to identify a subpopulation of cells with differentiation ability. In all these years there is an ongoing debate trying to identify “truly” progenitor cells in the breast; therefore any study which works in that direction is interesting, as it can provide further evidence for the presence of a mammary hierarchy. I only have minor comments for this paper.	We are very pleased that the reviewer finds our study interesting and supports our topic of research, and we thank the reviewer for a very thorough review and helpful suggestions.
- Line 55: the authors describe their previous paper, where TDLU-derived MEPs can give rise to K19+ or K19- cells. However, they fail to mention this is age-dependant. This should be included to avoid giving the impression of a generic behaviour of the cells during the totality of the life of the individual.	We thank the reviewer for the observant comment. In the revised version of the manuscript, we now have included age-dependency on p. 3, l. 55: “Whereas ductal MEP progenitors homogeneously differentiate into K19+ cells, TDLU-derived MEP are multipotent and generate both K19+ and K19- luminal cells as is also seen in an age-dependent manner in TDLUs in vivo¹⁰.”
- Samples are taken from reduction mammoplasty, which is often skewed by different BMIs. Are the BMI and the age of these women of significant difference? High BMI has been shown to have a protective effect on younger women, and a detrimental effect on older individuals. I understand the authors may not have this information, but it would be good practice to include it when analysing reduction mammoplasty. However, considering the difference in heterogeneity which is seen in the TDLU according to different ages (as per their previous publication), at least the age, if not the BMI, should be reported here, and differences in ages on the results (if any difference is present) should be described and discussed.	We agree with the reviewer that age and BMI cause biopsy-dependent variability in tissue from reduction mammoplasties. Unfortunately, the age of the donors at the time of surgery is the only information available to us. For immunohistochemistry and FACS of CD200, we included samples from women ranging from 13 to 59 years of age. We have now included this information in the methods part on p. 15, l. 324: “Normal breast tissue was obtained from 37 women aged between 13 and 59 years undergoing reduction mammoplasty for cosmetic reasons.” Importantly, however, in the present study, we did not observe that donor’s age influenced the results.
- Related to this, for scRNA sequencing, samples from 18 year old patients were chosen, but it is not explained why this particular age was chosen.	We chose samples from young women to avoid introducing additional variation due to parital status. Statistically, it is most likely for women in Denmark

	under the age of 21 to be nulliparous. We included this explanation in the Methods part on p. 16, l. 345-347 “While the parital status of the donors is unknown, biopsies from young donors at the age of 18 were selected to reduce the risk of introducing parity-related variability between samples.”
- Smooth muscle actin alpha is canonically abbreviated to either ACTA2 or SMA. I would suggest replacing either of these with sm actin, which is a very unusual abbreviation. Also, in most of the text, the protein is abbreviated, but then occasionally, such as in line 138, the full name appears again.	We appreciate the reviewer’s suggestion and abbreviated smooth muscle actin as “SMA”. Please note that we used “ACTA2” instead of “SMA” when referring to the gene instead of the protein.
- I am not an expert in cluster analysis, so I apologise in advance for this comment, if not relevant. It is not clear to me the difference between Suppl Table 2 and Suppl Table 3. By reading the table legend, I thought the latter was only a subset of the first, but I have noticed that the data is different between the two tables, for the same genes within the same clusters. Could you please improve the figure legend so it is clear the difference between the two set of data, or how the data is derived from the previous ones? Is the further division into anatomic regions the reason for the difference? PRDX1 1.54E-231 0.438438467 0.939 0.819 3.68E-227 0 PRDX1 6.93E-222 0.429187968 0.939 0.823 1.65E-217 0	We apologize for the lacking clarification between Supplementary Tables 2 and 3. Supplementary Table 2 shows a list of differentially expressed genes (DEGs) between all ten clusters. In contrast, Supplementary Table 3 shows DEGs of only clusters 0-6, because we excluded clusters 7, 8, and 9 from the analysis. DEGs are calculated by comparing gene expression in one cluster compared to the residual clusters. After exclusion of three clusters, the average gene expression levels in the “residual” clusters changed. Therefore, the values in Supplementary Table 3 are different from Supplementary Table 2. In the revised version of the manuscript, we included this information in the figure legend of Supplementary Table 3: “Tables display gene names (gene), p value (p_val), average logarithmic fold changes of cells per cluster (avg_logFC), percentages of cells expressing the gene in the cluster (pct.1) versus in all other clusters (pct.2) after excluding clusters 7, 8, and 9 from the analysis, the adjusted p value after correction for multiple testing (p_val_adj), and the cluster for A) all DEGs in clusters 0 to 6, B) DEGs encoding surface proteins in clusters 0 to 6, C) DEGs in TDLUs, and D) DEGs in ducts.”
- Line 105: The authors screened several antibodies and determined that CD200 was “superior” to AMIGO2. How did they defined “superiority”? Which outcomes were they looking for after IHC or FACS (I guess specificity, or intensity?). The authors would need to explain this in the methods, and possibly include a supplementary figure with at least a couple of	We thank the reviewer for bringing this to our attention. The antibody against AMIGO2 that we employed did not give any signal neither in immunohistochemistry nor in FACS experiments under the conditions tested. The sentence has now been changed to “A screen of antibodies revealed that CD200, but not AMIGO2, was suitable for both immunostaining and FACS”. (p. 6, l. 106)

representative images to allows us to understand their choice of antibody/protein.	
- Line 108: depending on the biopsy, the percentage of CD200^{low} cells accounts to a different percentage of MEP. How would the authors explain this difference?	Human tissue exhibits a great degree of interpersonal variation, which can be caused by multiple factors including age, parity, BMI, and menstrual cycle phase. For this reason, we repeated the experiments involving primary tissue such as FACS of CD200 cells using numerous different biopsies, and we deliberately included data to illustrate this variation. The only available information about the donors was their age at the time of surgery. However, we did not observe any correlation of the differences in the percentage of CD200^{low} cells with donor age.
- Line 119: Could the authors explain how was immunostaining (Fig 1C) quantified in the methods? Also, the choice of violin plot in Fig 1B is good, but the overlay with all the data points makes it hard to read. I would suggest the authors to remove the individual dots and just include the n numbers near the x axis. Finally, was immunostaining also tested for K14? If not, the text in line 120 should be edited to: "This was confirmed for K17 by immunostaining". I would also recommend to use the same nomenclature in the figure, so either K17 or KRT17. Maybe the use of K17/KRT17 could be adopted, if the authors want to use two terms.	We appreciate the helpful suggestions. We think the reviewer may be referring to Figure 2C instead of 1C? A set of 33 biopsies was evaluated twice independently by a blinded assessor who assigned a score to each biopsy based on number of positively stained cells and staining intensity. We agree that the violin plots in Figure 2B look crowded. However, the graph guidelines of Nature journals ask to include all data points individually in the graphs. Therefore, to increase readability, we replaced Figure 2B with violin plots with smaller data points.  We also tested K14 immunostaining, which gave very similar results to K17. However, since we do not show it here, we agree to change the text accordingly (p. 7, l. 119): "This was confirmed for K17 by immunostaining." We would like to keep both "K17" and "KRT17", since K17 refers to Keratin 17 protein, while KRT17 designates the gene.
- Line 123: "came out strongest" does not sound technical. Please edit to "the intensity of the signal of CD200 was strongest", or "the expression of CD200, as determined by immunostaining, was higher".	We thank the reviewer for this suggestion, and the text has been changed on p. 7, l. 122-124: "Accordingly, in a sample of biopsies, the expression of CD200, as determined by immunostaining, was higher in terminal ducts

	compared to alveoli in fifteen out of thirty biopsies (Supplementary Fig. 2B).
- Suppl Fig 2B. The bottom scheme is not very clear. I understand the comparison, but it is not clear what does the length of the line represent. Are those different biopsies? I think it is quite confusing, and should be either improved for clarity or visualised in a different way.	We thank the reviewer for drawing our attention to this. We realize that the lines in the figure were confusing and have now excluded them in the revised version of the manuscript.
- Line 142: please state how the expression was measured here (immunohistochemistry?) In relation to this, Fig 3A should include statistical analysis to determine the significant difference in the expression of actin in the last group compared to the others. Comparing the slope of decrease during passages could be a good way to measure this. Also, the variation between samples is so big that I would recommend using a boxplot rather than a barchart, as it would be more informative. Also, how was the induction of hypoxia tested after incubation with lower oxygen? Did the authors test the expression of markers such as HIF1a? This should be added in the method section.	SMA-positive cells were determined using the “CellCounter” function of the software Fiji. Only cells were included that were α-SMA⁺ on a K14⁺ background since the fibroblast feeder can acquire α-SMA expression. We thank the reviewer for the suggestion to use a box plot, and we have changed Figure 3A accordingly.  We did not test for induction of hypoxia in the cells. However, checking for induction of hypoxia is not common practice when keeping cells under hypoxic conditions as seen in numerous publications (Tretiach, Neuroscience Letters 2005; Lee, Biochem Biophys Res 2006; Liu, Microvascular Res 2008; Rosová, Stem Cells 2009; Ueyama, J Cell Mol Med 2012; Zhang, PLoS One 2013; Lee, Bone 2015)
- Line 174: The authors have transduced the cells with 3 constructs, subsequently: hTERT, shp53 and PIK3CA. They then observed elevated proliferation in both hTERT/shp53 and hTERT/shp53/PIK3CA - transduced cells. From the text, it does not look like the authors have cells with only hTERT/PIK3CA. I therefore wonder how they can explain their statement saying: “... the introduction of mutant PIK3CA does not increase	We thank the reviewer for this comment. This statement relates to the fact that overexpression of PIK3CA^{H1047R} does not increase proliferation compared to cells expressing hTERT-shp53 only (Supplementary Figure 5B). We clarified this by modifying the text as follows (p. 9, l. 176-177): “This implies that introduction of mutant PIK3CA does not increase the proliferative capacity compared to cells expressing hTERT-shp53 only.”

the proliferative capacity per se". Assuming that "per se" indicates without either hTERT, shp53, or both, proliferation data of cells transduced with PIK3CA or hTERT/PIK3CA should be shown before that statement is made.	
- Line 187: "remain faithful to their origins" - please edit with a more appropriate technical language	We appreciate this suggestion and made the following adjustment on p. 10, l. 188-190: "Notably, upon induced differentiation, both CD200^{low}-hTERT-shp53-PIK3CA^{H1047R} and CD200^{high}-hTERT-shp53-PIK3CA^{H1047R} maintain their original phenotypes and only CD200^{low}-hTERT-shp53-PIK3CA^{H1047R} generates mature luminal K14⁻/K19⁺ cells (Fig. 4C)."
- Line 266: remove the word keratin, since the abbreviation is present	We thank the reviewer for paying careful attention and have corrected both errors.
- Line 413: Invitrogen. Spelling mistake.	

REVIEWERS' COMMENTS:

Reviewer #1 (Remarks to the Author):

Authors have addressed concerns raised in the last review. However, abstract has to be modified to indicate that PIK3CA mutant overexpression studies were done in the context of p53 depletion.

Reviewer #2 (Remarks to the Author):

It appears that authors have address reviewers comments

Reviewer #3 (Remarks to the Author):

Thank you for addressing and implementing most of the reviewers' concerns. This is a very interesting study.